# COMBINING EUCLIDEAN AND HYPERBOLIC REPRESENTATIONS FOR NODE-LEVEL ANOMALY DETECTION

## ABSTRACT

Node-level anomaly detection (NAD) is challenging due to diverse structural patterns and feature distributions. As such, NAD is a critical task with several applications which range from fraud detection, cybersecurity, to recommendation systems. We introduce Janus, a framework that jointly leverages Euclidean and Hyperbolic Graph Neural Networks to capture complementary aspects of node representations. Each node is described by two views, composed by the original features and structural features derived from random walks and degrees, then embedded into Euclidean and Hyperbolic spaces. A multi Graph-Autoencoder framework, equipped with a contrastive learning objective as regularization term, aligns the embeddings across the Euclidean and Hyperbolic spaces, highlighting nodes whose views are difficult to reconcile and are thus likely anomalous. Experiments on four real-world datasets show that Janus consistently outperforms shallow and deep baselines, empirically demonstrating that combining multiple geometric representations provides a robust and effective approach for identifying subtle and complex anomalies in graphs. We publicly release our source code at
`https://anonymous.4open.science/r/JANUS-5EDF/`.

## 1 INTRODUCTION

Anomaly detection plays a crucial role in machine learning, with impactful use cases in cybersecurity, social media analysis, financial fraud prevention, and healthcare (Akoglu et al., 2015; West & Bhattacharya, 2016). Graph anomaly detection extends this paradigm to domains where data naturally exhibits relational structure and can be modeled as a graph. Within this setting, node-level anomaly detection methods (Ma et al., 2021) focus on identifying anomalous nodes, where the notion of "anomaly" varies depending on the application. In general, anomalous nodes are those that deviate substantially from the majority in terms of their attributes or structural connectivity.

There exists a vast body of literature on graph anomaly detection, ranging from tree-based methods to deep learning frameworks. Supervised approaches (Tang et al., 2023) rely on labeled anomalies; however, such labels are often scarce or unavailable, and these methods are inherently restricted to detecting only known types of anomalies. To overcome this limitation, alternative strategies employ unsupervised (Liu et al., 2022) or semi-supervised (Ioannidis et al., 2019) paradigms. Graph Neural Networks (GNNs) have emerged as the state of the art in graph representation learning, and have been extensively applied to NAD. Among the most common GNN-based strategies are graph autoencoders (Ding et al., 2019; Fan et al., 2020; Roy et al., 2024) and graph contrastive learning (Liu et al., 2021a; Pan et al., 2023). Autoencoder-based methods learn to embed nodes and edges into a latent space and then reconstruct the original features and adjacency matrix. Nodes with high reconstruction error are considered anomalous, as their structure or features deviate significantly from the majority. Contrastive learning approaches, on the other hand, enforce representation similarity between positive pairs (e.g., nodes from the same neighborhood or different augmentations of the same node) while pushing apart negative pairs (e.g., nodes from different neighborhoods). Different works vary in how positive and negative pairs are defined, but the core principle remains to highlight anomalies as those nodes that fail to align with learned similarity patterns. Other approaches, such as Luo et al. (2022), exploit both methods.

Current state-of-the-art models adopt classic GNNs that map nodes and edges in an Euclidean latent space. Lately, scholars are exploring non-Euclidean geometries such as Hyperbolic. Such space,

characterized by its constant negative curvature, diverges from the flatness of Euclidean geometry. The hyperboloid manifold is often favored for its numerical stability (Fu et al., 2024) and for its capability in preserving hierarchical relationships (Nickel & Kiela, 2018). Some approaches tried to combine different spaces to enhance their benefits (Iyer et al., 2022; Gu et al., 2018; Cho et al., 2023).

In this regard, we aim at improving the detection capabilities of NAD models by adopting a multi-geometry hybrid framework composed of a Graph-Autoencoder equipped with a contrastive learning objective. The idea is to: (i) combine spaces with diverse metric structures, i.e., Euclidean and Hyperbolic latent spaces, to highlight different complementary structural and semantic features that may be emphasized differently depending on the underlying geometry; and (ii) adopt one GAE for each geometry, eventually combining the different node representations to reconstruct the original features and to align different views of the same node, accordingly to the Graph Autoencoder and Graph Contrastive Learning approaches, respectively. To the best of our knowledge, this is the first attempt at doing this.

To summarize, our main contributions are threefold:

- We propose Janus, a novel multi-geometry Graph Autoencoder augmented with a contrastive learning objective, specifically designed for node-level graph anomaly detection.

- We introduce, to the best of our knowledge, the first framework that jointly exploits Euclidean and Hyperbolic latent spaces for node-level graph anomaly detection, thereby capturing complementary structural and semantic patterns that are overlooked by single-geometry models.

- We conduct an extensive experimental evaluation across four real-world datasets, demonstrating that Janus consistently outperforms state-of-the-art baselines.

The paper is organized as follows. Section 2 reviews existing work on node-level graph anomaly detection. Section 3 introduces the necessary background on hyperbolic and hybrid geometric spaces. Section 4 details the proposed method. Section 5 presents the empirical evaluation against baseline models. Finally, Section 6 concludes the manuscript.

## 2 RELATED WORKS

**Node-Level Anomaly Detection with GNNs.** A rich body of work explores node-level anomaly detection on graphs using GNNs. Early methods like ANOMALOUS (Peng et al., 2018) jointly learned representations and selected important attributes via CUR decomposition to mitigate noisy features. Reconstruction-based approaches, such as DOMINANT (Ding et al., 2019), AnomalyDAE (Fan et al., 2020), Radar (Li et al., 2017), and GAD-NR (Roy et al., 2024), leverage GNN/autoencoder frameworks to detect anomalies via high reconstruction error in either structure or attributes.

Contrastive and generative techniques refine this paradigm. GRADATE (Duan et al., 2023a) and NLGAD (Duan et al., 2023b) employ multi-scale contrastive learning—subgraph vs. subgraph, node vs. context—to learn robust normality representations that highlight deviations. PREM (Pan et al., 2023), with a lightweight preprocessing-and-matching pipeline, achieves competitive detection efficiency and simplicity. SmoothGNN (Dong et al., 2025) identifies smoothing patterns (ISP/NSP) to isolate anomalous representations explicity. Truncated Affinity Maximization (Qiao & Pang, 2023) introduces one-class homophily modeling: it iteratively truncates non-homophilic edges to learn tight affinity for normal nodes. Other works aim at improving the neighbor selection, such as Reinforcement Neighborhood Selection (RAND) (Bei et al., 2023), using RL to improve anomaly scoring.

**Hyperbolic Graph Neural Networks.** Hyperbolic geometry offers, Krioukov et al. (2010), a powerful way to represent the hierarchical structures found in complex networks, Clauset et al. (2008). Hyperbolic GNNs leverage hyperbolic geometry to effectively model hierarchical and scale-free structures in graphs. Though their application to node-level anomaly detection remains unexplored, recent works highlight their potential. For example, Gu & Zou (2024) revisits anomaly detection through the lens of hyperbolic neural networks, showing improved detection. Comparative studies (Touahria Miliani et al., 2024) in cybersecurity benchmark Poincaré and Lorentz embedding models, demonstrating enhanced anomaly separability under hyperbolic distances. At the graph level,

HC-GLAD (Fu et al., 2024) employs dual hyperbolic contrastive learning to distinguish anomalous graphs.

**Combination of different spaces.** Recent work has shown that embedding data in different spaces can improve representation quality. Iyer et al. (2022) introduces a dual-geometry embedding scheme that assigns different portions of a knowledge graph to different geometric spaces, demonstrating improved modeling of two-view KGs by explicitly separating regions with distinct geometric priors. Gu et al. (2018) proposes learning embeddings in a mixed space composed of Spherical, Euclidean, and Hyperbolic components, jointly learning both the embeddings and the curvature of each component. Similarly, in Cho et al. (2023) the authors propose to apply such a procedure to the attention head of Graph Transformer, allowing the model to learn appropriate curvatures and exploit Euclidean, hyperbolic, and spherical components within a unified, attention-based encoder. The idea of using multiple curvature experts has been explored recently in HELM (He et al., 2025). The authors propose training Large Language Models (LLMs) components that operate in distinct curvature spaces (a Mixture-of-Curvature Experts), showing that combining curvature-specialized experts can capture richer semantic hierarchies than single-geometry models in LLMs.

In this work, we extend this line of research to node-level anomaly detection, introducing a contrastive, autoencoder-based framework that jointly exploits Euclidean and hyperbolic representations. To the best of our knowledge, this is the first study to integrate different geometries in this task.

## 3 GEOMETRIC PRELIMINARIES

For the reader's convenience, we will briefly summarize a few theoretical concepts that will be central to our implementation of computations in the hyperbolic setting and the comparison of embeddings in different metric spaces.

**Product metric spaces.** For relevant definitions related to metric spaces, the reader may refer to (Deza & Deza, 2016). Here, we'll collect a few useful facts for what follows. Given a metric space $(M, d)$ and $k > 0$, the new function

$$d'(x, y) = k \frac{d(x, y)}{1 + d(x, y)} \tag{1}$$

for $x, y \in M$ is again a distance on $M$ whose diameter is less then $k$, namely $0 \le d' < k$, see (Deza & Deza, 2016, p. 88). Furthermore, if $(M_1, d_1)$, $(M_2, d_2)$ are two metric spaces, the new function

$$d''((x_1, y_1), (x_2, y_2)) = d_1(x_1, y_1) + d_2(x_2, y_2) \tag{2}$$

for all $(x_1, y_1), (x_2, y_2) \in M_1 \times M_2$, it is a distance on $M_1 \times M_2$, see (Deza & Deza, 2016, p. 93). Fix $k_1, k_2 > 0$, it follows from Equations 1, 2 that the new function

$$d'''((x_1, y_1), (x_2, y_2)) = k_1 \frac{d_1(x_1, y_1)}{1 + d_1(x_1, y_1)} + k_2 \frac{d_2(x_2, y_2)}{1 + d_2(x_2, y_2)} \tag{3}$$

is a distance on $M_1 \times M_2$.

**Hyperbolic Setting.** As a model of hyperbolic geometry, we choose the Minkowski hyperboloid (of constant negative curvature $-1$) because it offers relative numerical stability and streamlines implementation, similar to prior works Fu et al. (2024); Gu & Zou (2024). First of all, consider coordinates $(x_0, \ldots, x_d)$ of $\mathbb{R}^{d+1}$ and define the Minkowski inner product as $\langle \mathbf{x}, \mathbf{y} \rangle_{\mathcal{L}} = -x_0 y_0 + \sum_{i=1}^{d} x_i y_i$ for vectors $\mathbf{x}, \mathbf{y} \in \mathbb{R}^{d+1}$, then The $d$-dimensional hyperboloid model is defined as:

$$\mathbb{H}^d = \left\{ \mathbf{x} \in \mathbb{R}^{d+1} \mid \langle \mathbf{x}, \mathbf{x} \rangle_{\mathcal{L}} = -1, \ x_0 > 0 \right\} \tag{4}$$

and the geodesic distance between two points $\mathbf{x}, \mathbf{y} \in \mathbb{H}^d$ is given by:

$$d_{\mathcal{L}}(\mathbf{x}, \mathbf{y}) = \text{arcosh}\left( -\langle \mathbf{x}, \mathbf{y} \rangle_{\mathcal{L}} \right) . \tag{5}$$

At a point $\mathbf{x} \in \mathbb{H}^d$, we identify the tangent space $T_{\mathbf{x}} \mathbb{H}^d$ with the linear subspace:

$$T_{\mathbf{x}} \mathbb{H}^d = \left\{ \mathbf{v} \in \mathbb{R}^{d+1} \mid \langle \mathbf{x}, \mathbf{v} \rangle_{\mathcal{L}} = 0 \right\} \tag{6}$$

As is common in literature, the passage from $\mathbb{H}^d$ to a tangent space is the main technical point for generalising ordinary Euclidean operations (e.g., vector-matrix multiplication) to hyperbolic settings. (See Section for more details.)

The tool that provides the mapping from a tangent space to the hyperboloid is the so-called Exponential Map. The mapping in the opposite direction is performed by the Logarithmic Map, which is formally defined as follows:

- *Exponential Map*: fix $\mathbf{x} \in \mathbb{H}^d$, $\exp_{\mathbf{x}} \colon T_{\mathbf{x}}\mathbb{H}^d \to \mathbb{H}^d$ it is defined as

$$\exp_{\mathbf{x}}(\mathbf{v}) = \cosh(\|\mathbf{v}\|_{\mathcal{L}})\mathbf{x} + \sinh(\|\mathbf{v}\|_{\mathcal{L}})\frac{\mathbf{v}}{\|\mathbf{v}\|_{\mathcal{L}}} \tag{7}$$

where $\mathbf{v} \in T_{\mathbf{x}}\mathbb{H}^d$ and $\|\mathbf{v}\|_{\mathcal{L}} = \sqrt{\langle \mathbf{v}, \mathbf{v} \rangle_{\mathcal{L}}}$.

- *Logarithmic Map*: fix $\mathbf{x} \in \mathbb{H}^d$, $\log_{\mathbf{x}} \colon \mathbb{H}^d \to T_{\mathbf{x}}\mathbb{H}^d$ is defined as

$$\log_{\mathbf{x}}(\mathbf{y}) = d_{\mathcal{L}}(\mathbf{x}, \mathbf{y}) \cdot \frac{\mathbf{y} + \langle \mathbf{x}, \mathbf{y} \rangle_{\mathcal{L}}\mathbf{x}}{\|\mathbf{y} + \langle \mathbf{x}, \mathbf{y} \rangle_{\mathcal{L}}\mathbf{x}\|_{\mathcal{L}}} \tag{8}$$

where $\mathbf{y} \in \mathbb{H}^d$.

As is rather customary, to simplify computations and streamline implementation, see for instance Fu et al. (2024), we fix the tangent space at $\mathbf{o} = (1, 0, \ldots, 0) \in \mathbb{H}^d$. Under this hypothesis, Equation 6 shows that the linear subspace $T_{\mathbf{o}}\mathbb{H}^d$ is defined by the equation $x_0 = 0$. Consequently, if $\mathbf{v} \in T_{\mathbf{x}}\mathbb{H}^d$, then $\mathbf{v} = [0, \mathbf{v}^E]$, meaning $\mathbf{v}$ can be written as the concatenation of $0$ and a vector $\mathbf{v}^E \in \mathbb{R}^d$. Furthermore, $\langle \mathbf{v}, \mathbf{v} \rangle_{\mathcal{L}} = \langle \mathbf{v}^E, \mathbf{v}^E \rangle$ and $\|\mathbf{v}\|_{\mathcal{L}} = \|\mathbf{v}^E\|_2$, where $\langle \cdot, \cdot \rangle$ and $\|\cdot\|_2$ denote the usual scalar product and the associated $L_2$ norm in $\mathbb{R}^d$. In light of this discussion, if $\mathbf{v} \in T_{\mathbf{o}}\mathbb{H}^d$, we can write the exponential map as:

$$\exp_{\mathbf{o}}(\mathbf{v}) = \exp_{\mathbf{o}}\left([0, \mathbf{v}^E]\right) = \left(\cosh(\|\mathbf{v}^E\|_2), \sinh(\|\mathbf{v}^E\|_2) \times \frac{\mathbf{v}^E}{\|\mathbf{v}^E\|_2}\right). \tag{9}$$

## 4 Approach

**Problem Statement.** Given an undirected graph $G = (X, E)$, where $X \in \mathbb{R}^{n \times d}$ represents the node features and $E$ denotes the set of edges, we assume the existence of a subset $\hat{X} \subseteq X$ corresponding to anomalous nodes. We further assume to have the set $Y = \{y_0, \ldots, y_n\}$, where $y_i = 1$ if $x_i \in \hat{X}$, and $y_i = 0$ vice versa. The goal is to estimate the probability distribution $P(y|G)$, where $y \in \{0, 1\}$ indicates whether the node $x$ is anomalous. To this end, we aim to learn a function $f(G) \to Y$, where $f$ assigns an anomaly score to each node $x \in X$, reflecting the likelihood of being anomalous.

**Janus Framework.** To address the aforementioned problem, we introduce a multi-graph autoencoder framework, equipped with a contrastive learning objective. Specifically, we construct two distinct views for each node, denoted by $X^s$ and $X^g$, and apply a mapping function $f$ that projects nodes into a latent space. The function $f$ maps similar node views close to each other in this space. Consequently, we expect that $d(x_i^s, x_i^g) > d(x_j^s, x_j^g)$ if $y_i = 1$ (anomalous) and $y_j = 0$ (normal), which reflects the difficulty of $f$ in aligning representations of anomalous node views due to their inherent dissimilarity.

**Node views.** First, we define the node $i$ views as $x_i^s = x_i \in X$, hence the node starting features, and $x_i^g = [RW_i \| D_i]$ as a combination of local and global structural features (Liu et al., 2023), respectively. In particular, $D_i$ represents the one-hot encoding of node degrees (Qiu et al., 2020; Xu et al., 2019), while $RW_i$ (Dwivedi et al., 2022) is computed as follows.

$$RW_i = [T_{ii}, T_{ii}^2, \ldots, T_{ii}^{d_{rw}}] \in \mathbb{R}^{d_{rw}} \tag{10}$$

$$T = \tilde{A}D^{-1} \tag{11}$$

Where $A$ is the adjacency matrix of $G$ and $\tilde{A} = A + I$ is the adjacency matrix with added self-connections, and $D$ is the corresponding degree matrix.

**Mapping function.** Graph Neural Networks (GNNs) have been extensively demonstrated to be effective for graph representation learning tasks. Traditional GNNs embed nodes into a Euclidean

latent space, a geometric space where Euclidean distance and other classical geometric properties hold. More recently, Hyperbolic Graph Neural Networks (HGNNs) (Wang et al., 2021a; Liu et al., 2021b; Peng et al., 2021; Bai et al., 2023) have gained attention for their ability to model hierarchical and complex structures, achieving promising results across various domains.

In our approach, we leverage both Euclidean and non-Euclidean (hyperbolic) GNNs, denoted respectively as $\text{GNN}^e$ and $\text{GNN}^h$. The key idea is to represent nodes in multiple geometric spaces, thereby capturing complementary structural and semantic features that may be emphasized differently depending on the underlying geometry. In particular, we adopt an autoencoder-like architecture consisting of $\text{GNN}^e_{enc}$ and $\text{GNN}^h_{enc}$ for encoding, and $\text{GNN}^e_{dec}$ and $\text{GNN}^h_{dec}$ for decoding.

To formally characterize the GNNs employed in our approach, we adopt the standard Graph Convolutional Network formulation. Nonetheless, the derivation extends naturally to any GNN model.

**Euclidean GNN.** $\text{GNN}^e_{enc}$ generates node embeddings through a message-passing architecture, which aggregates neighbors node features. The formula for generating the set of nodes embeddings $H \in \mathbb{R}^{n \times k}$ for the l-th layer is described as follows.

$$H_{(l+1)} = \sigma \left( \tilde{D}^{-1/2} \tilde{A} \tilde{D}^{-1/2} H_{(l)} W_{(l)} \right) \tag{12}$$

Where $H_{(0)} = X$, $W_{(l)} \in \mathbb{R}^{D_l \times D_{l+1}}$ is a trainable weight matrix for layer $l$. In particular, we define as $H^s_{(l+1)}$ and $H^g_{(l+1)}$ when $H^s_{(0)} = X^s$ and $H^s_{(0)} = X^s$, respectively.

**From Euclidean GNN to Hyperbolic GNN.** Euclidean GNNs use standard vector operations and inner products in $\mathbb{R}^d$. Hyperbolic GNNs map data in $\mathbb{H}^d$ and use exponential or logarithmic mappings to move between the manifold and tangent spaces. This allows them to perform hyperbolic versions of the corresponding Euclidean operations. In particular, following Equations 9 and 7, we map each $x^s \in X^s$ in the hyperbolic model $\mathbb{H}^d$:

$$\hat{x}^s = \exp_{\mathbf{o}} \left( [0, x^s] \right) \tag{13}$$

The same logic applies for $x^g \in X^g$. We denote $\hat{X}^s = \{\hat{x}^s_0, \ldots, \hat{x}^s_n\}$ and $\hat{X}^g = \{\hat{x}^g_0, \ldots, \hat{x}^g_n\}$ as the sets of node features in the different views mapped in the hyperbolic model.

The graph convolutions in Hyperbolic GNNs, namely $\text{GNN}^h_{enc}$, are applied within the tangent space. As such, $\hat{X}^s$ and $\hat{X}^g$ are first mapped into the tangent space using Equation 8, then we can apply the HGNN convolution:

$$\hat{H}_{(l+1)} = \sigma \left( \tilde{D}^{-1/2} \tilde{A} \tilde{D}^{-1/2} \hat{H}_{(l)} W_{(l)} \right) \tag{14}$$

Similarly to Equation 12, we define as $\hat{H}^s_{(l+1)}$ and $\hat{H}^g_{(l+1)}$ when $\hat{H}^s_{(0)} = \hat{X}^s$ and $\hat{H}^s_{(0)} = \hat{X}^s$, respectively.

**Contrastive Learning with Product Metric.** Following prior works, we adopt a node-level contrastive loss which maximizes the agreement between different views of the same node both in Euclidean and Hyperbolic spaces. In particular, the loss function can be formalized as follows.

$$\mathcal{L}_{cl} = \frac{1}{2n} \sum_{i=0}^{n} l_1 \left( h^g_i, \hat{h}^g_i, h^s_i, \hat{h}^s_i \right) + l_2 \left( h^g_i, \hat{h}^g_i, h^s_i, \hat{h}^s_i \right) \tag{15}$$

$$l_1 \left( h^g_i, \hat{h}^g_i, h^s_i, \hat{h}^s_i \right) = - \log \frac{e^{\left( -\mathcal{D}(h^g_i, \hat{h}^g_i, h^s_i, \hat{h}^s_i)/\tau \right)}}{\sum_{j \in \{0,\ldots,n\},\, j \neq i} e^{\left( -\mathcal{D}(h^g_i, \hat{h}^g_i, h^s_j, \hat{h}^s_j)/\tau \right)}} \; . \tag{16}$$

$$l_2 \left( h^s_i, \hat{h}^s_i, h^g_i, \hat{h}^g_i \right) = - \log \frac{e^{\left( -\mathcal{D}(h^s_i, \hat{h}^s_i, h^g_i, \hat{h}^g_i)/\tau \right)}}{\sum_{j \in \{0,\ldots,n\},\, j \neq i} e^{\left( -\mathcal{D}(h^s_i, \hat{h}^s_i, h^g_j, \hat{h}^g_j)/\tau \right)}} \; . \tag{17}$$

In particular, in our approach, given four quantites $\{\alpha_1, \alpha_2, \alpha_3, \alpha_4\}$, we define the function $\mathcal{D}$ using Equation 18 between views in Euclidean and Hyperbolic spaces:

$$\mathcal{D}(\alpha_1, \alpha_2, \alpha_3, \alpha_4) = \frac{1}{2}\left(\frac{d(\alpha_1, \alpha_3)}{1 + d(\alpha_1, \alpha_3)} + \frac{\hat{d}(\alpha_2, \alpha_4)}{1 + \hat{d}(\alpha_2, \alpha_4)}\right) \tag{18}$$

where $d$ is the $L_2$ norm and $\hat{d}$ is the geodesic distance in the Hyperbolic space defined in Equation 5. Observe that the right hand side in parentheses of the above formula is bounded by 2 we then divide by two to normalize $\mathcal{D}$.

**Decoder Module.** After generating the node embeddings both in Euclidean and Hyperbolic spaces, we aim at reconstructing the input graph $G$, thus its adjacency matrix $A$ and the original and structural node features $X^s$ and $X^g$, respectively. As prior work, we generate the reconstructed adjacency matrix $\mathcal{A}$ for original and structural features as follows.

$$\mathcal{A}^s = sigmoid(H^s H^{s\top}) \tag{19}$$

$$\mathcal{A}^g = sigmoid(H^g H^{g\top}) \tag{20}$$

Similarly, for the Hyperbolic embeddings:

$$\hat{\mathcal{A}}^s = sigmoid(\hat{H}^s \hat{H}^{s\top}) \tag{21}$$

$$\hat{\mathcal{A}}^g = sigmoid(\hat{H}^g \hat{H}^{g\top}) \tag{22}$$

Hence, we can define the reconstruction loss for the adjacency matrix:

$$\mathcal{L}_{adj} = ||A - \mathcal{A}^s||^2 + ||A - \mathcal{A}^g||^2 + ||A - \hat{\mathcal{A}}^s||^2 + ||A - \hat{\mathcal{A}}^s||^2 \tag{23}$$

For reconstructing node features, we adopt two GNNs which we define as $\text{GNN}_{dec}^e$ and $\text{GNN}_{dec}^h$ for Euclidean and Hyperbolic embeddings:

$$\mathcal{H}_{(l+1)} = \sigma\left(\tilde{D}^{-1/2}\tilde{A}\tilde{D}^{-1/2}\mathcal{H}_{(l)}W_{(l)}\right) \tag{24}$$

$$\hat{\mathcal{H}}_{(l+1)} = \sigma\left(\tilde{D}^{-1/2}\tilde{A}\tilde{D}^{-1/2}\hat{\mathcal{H}}_{(l)}W_{(l)}\right) \tag{25}$$

Here, we define the reconstructed starting and generated node features as $\mathcal{H}^s$ and $\mathcal{H}^g$, respectively. The result of Eq 24 is $\mathcal{H}^s$ whether $\mathcal{H}_{(0)}^s = H^s$, and $\mathcal{H}^g$ whether $\mathcal{H}_{(0)}^g = H^g$. The same applies for $\hat{\mathcal{H}}^s$ and $\hat{\mathcal{H}}^g$.

For computing the node features reconstruction error in Euclidean and Hyperbolic spaces, namely $X^s, X^g, \hat{X}^s, \hat{X}^g$, we adopt, similar to Eq. 18, the product metric formula to combine distances:

$$\mathcal{L}_{node} = \frac{1}{2} \times \left(\frac{d(X^g, \mathcal{H}^g) + d(X^s, \mathcal{H}^s)}{1 + d(X^g, \mathcal{H}^g) + d(X^s, \mathcal{H}^s)} + \frac{\hat{d}(\hat{X}^g, \hat{\mathcal{H}}^g) + \hat{d}(\hat{X}^s, \hat{\mathcal{H}}^s)}{1 + \hat{d}(\hat{X}^g, \hat{\mathcal{H}}^g) + \hat{d}(\hat{X}^s, \hat{\mathcal{H}}^s)}\right) \tag{26}$$

Finally, combining all the components we define the loss function as follows.

$$\mathcal{L} = \mathcal{L}_{cl} + \lambda_1\mathcal{L}_{adj} + \lambda_2\mathcal{L}_{node} \tag{27}$$

Eq. 27 defines the loss function used to train the model, which can also serve as an anomaly scoring mechanism. For this purpose, we omit the adjacency reconstruction term by setting $\lambda_1 = 0$.

The complete workflow presented in this section is summarized in Algorithm 1.

Specifically, in our implementation, we employ uniform sampling to construct node neighborhoods, consistent with the strategy introduced in GraphSAGE (Hamilton et al., 2017). The neighborhood size is considered a tunable hyperparameter, and its optimal value is selected according to the characteristics of each dataset. For the underlying architecture of the employed GNNs, we adopt the Graph Isomorphism Network (GIN) (Xu et al., 2019), given its expressive power in distinguishing graph structures.

---

**Algorithm 1** Euclidean–Hyperbolic Node Anomaly Detection

---

**Require:** Graph $G = (V, E)$, adjacency $A$, node features $X$
**Ensure:** Anomaly scores $S$ for all nodes
1: $X^s \leftarrow X$ {Original node features}
2: Initialize $X^g \leftarrow []$
3: **for** each node $i \in V$ **do**
4: $\quad RW_i \leftarrow ComputeRandomWalkFeatures(A, i)$
5: $\quad D_i \leftarrow OneHotDegree(A, i)$
6: $\quad X^g[i] \leftarrow concatenate(RW_i, D_i)$
7: **end for**
8: $H^s, H^g \leftarrow \text{GNN}^e_{enc}(X^s, X^g, A)$ {Euclidean embeddings}
9: $\hat{X}^s \leftarrow ExpMap(X^s)$
10: $\hat{X}^g \leftarrow ExpMap(X^g)$
11: $\hat{H}^s, \hat{H}^g \leftarrow \text{GNN}^h_{enc}(\hat{X}^s, \hat{X}^g, A)$ {Hyperbolic embeddings}
12: $\mathcal{L}_{cl} \leftarrow ContrastiveLoss(H^s, H^g, \hat{H}^s, \hat{H}^g)$
13: $\mathcal{A}^s, \mathcal{A}^g \leftarrow AdjReconstruction(H^s, H^g)$
14: $\hat{\mathcal{A}}^s, \hat{\mathcal{A}}^g \leftarrow AdjReconstruction(\hat{H}^s, \hat{H}^g)$
15: $\mathcal{H}^s, \mathcal{H}^g \leftarrow \text{GNN}^e_{dec}(H^s, H^g)$
16: $\hat{\mathcal{H}}^s, \hat{\mathcal{H}}^g \leftarrow \text{GNN}^h_{dec}(\hat{H}^s, \hat{H}^g)$
17: $\mathcal{L}_{adj} \leftarrow AdjacencyLoss(A, \mathcal{A}^s, \mathcal{A}^g, \hat{\mathcal{A}}^s, \hat{\mathcal{A}}^g)$
18: $\mathcal{L}_{node} \leftarrow FeatureLoss(H^s, H^g, \mathcal{H}^s, \mathcal{H}^g, \hat{H}^s, \hat{H}^g, \hat{\mathcal{H}}^s, \hat{\mathcal{H}}^g)$
19: $\mathcal{L} \leftarrow \mathcal{L}_{cl} + \lambda_1 \mathcal{L}_{adj} + \lambda_2 \mathcal{L}_{node}$
20: **if** AnomalyDetectionMode **then**
21: $\quad \lambda_1 \leftarrow 0$ {Ignore adjacency reconstruction}
22: $\quad S \leftarrow \mathcal{L}$ {Use loss as anomaly score}
23: **end if**

---

## 5 EXPERIMENTS

This section provides a comprehensive empirical evaluation of the proposed model on four diverse real-world datasets. To gain a deeper understanding of its effectiveness, we investigate its behavior through the following guiding research questions:

**RQ1** How does Janus compare to state-of-the-art anomaly detection methods?

**RQ2** How do the different architectural components of Janus contribute to its final performance?

### 5.1 EXPERIMENTAL SETUP

Here we further detail how we devised the experiments in terms of datasets, competitors and evaluation protocols.

**Datasets.** For our empirical evaluation of Janus, we consider six real-world datasets: Disney Sánchez et al. (2013), Books Sánchez et al. (2013), Reddit Kumar et al. (2019); Wang et al. (2021b), T-Finance Tang et al. (2022), ACM Lv et al. (2021), and Flickr Zeng et al. (2019). A key characteristic of Disney, Books, Reddit and T-Finance datasets is that they contain naturally occurring (organic) anomalies, which better reflect the complexity of real-world environments. This differs from common benchmark practices where synthetic anomalies are generated by injecting artificial perturbations into clean graphs, such as Cora Sen et al. (2008) dataset. While convenient, such anomalies are often trivially separable from normal patterns and fail to capture the subtleties of real-world anomaly detection. Nonetheless, for providing a comprehensive overview, we also adopt ACM and Flickr datasets, whose anomalies are synthetically injected. In Table 1 we report dataset statistics, including numbers of nodes, edges, features, and anomaly ratios.

**Competitors.** To benchmark the performance of Janus, we compare it against six widely used anomaly detection methods, spanning from classical shallow approaches to recent deep graph-based

| Dataset | #Nodes | #Edges | #Features | %Anomaly Ratio |
|---------|--------|--------|-----------|----------------|
| Disney | 124 | 335 | 28 | 4.8 |
| Books | 1,418 | 3,695 | 21 | 2.0 |
| Reddit | 10,984 | 168,016 | 64 | 3.3 |
| T-Finance | 39,357 | 21,222,543 | 10 | 4.6 |
| ACM | 16,484 | 71,980 | 8,337 | 597 |
| Flickr | 7,575 | 239,738 | 12,074 | 450 |

Table 1: Statistics on the employed datasets including number of nodes, edges, features, and percentage of anomalous nodes

model: **LOF** Breunig et al. (2000), **Isolation Forest** Liu et al. (2008), **ANOMALOUS** Peng et al. (2018), **DOMINANT** Ding et al. (2019), **CONAD** Xu et al. (2022), **CARD** Wang et al. (2024). All baselines are implemented through the standardized PyGOD framework Liu et al. (2024; 2022). Further details are provided in the Appendix B.

**Evaluation Protocol.**   We assess anomaly detection performance using two well-established metrics: ROC-AUC and Average Precision (AP). The former quantifies the model's discriminative ability by considering both the true positive rate (correctly identified anomalies) and the false positive rate (normal nodes misclassified as anomalies). AP summarizes the Precision–Recall curve, reflecting the trade-off between precision and recall metrics across thresholds. A key advantage of both metrics is that they do not require fixing a decision threshold, which is crucial in anomaly detection tasks. Given the strong class imbalance typically observed in such settings, where anomalies represent only a small proportion of the nodes (Table 1), we use AP as the primary selection metric during hyperparameter tuning. To ensure robustness, we repeat each experiment using five random seeds. The reported results correspond to the mean performance together with the standard deviation, as summarized in the subsequent tables. We assess statistical significance of the results using the Wilcoxon signed-rank test with a confidence level of 90%. In the tables, the best-performing models are highlighted in bold, and when two or more models are statistically indistinguishable, all tied results are marked in bold.

Consistent with previous studies Ding et al. (2019); Xu et al. (2022); Wang et al. (2024), our evaluation is performed in a transductive setting. The complete list of hyperparameters employed for training Janus is presented in the Appendix in Table 6. Batch size and number of neighbors sampled are dataset-specific. For competitor models, we determine the best-performing configuration through empirical exploration of hyperparameters tailored to their architectures, specifically tuning hidden layer dimensions, dropout probabilities, and the trade-off coefficient balancing node feature and structural reconstruction losses.

## 5.2 RESULTS

To address **RQ1**, Table 2 presents the comparative performance of the proposed model against the considered baselines. The last row reports the relative percentage improvement with respect to the best-performing competitor, computed as the highest mean value for each dataset and metric. A timeout of 12 hours per seed was imposed for all experiments; under this constraint, CARD failed to complete training on T-Finance, ACM and Flickr datasets, for which we therefore report Out-Of-Time (OOT). Even using default parameters for CARD, it lead to OOT, which is caused by the multiple computations in its implementation (community detection, multiple augmentations, and large similarity computations). Overall, the proposed approach consistently achieves superior results across datasets, yielding relative improvements of up to $+32.4\%$ in ROC-AUC and $+337.5\%$ in AP with respect to the best competitor.

To further strengthen the evaluation of the proposed approach, we report in Figure 1 the Cumulative Gain curves across the considered datasets. These plots illustrate the percentage of correctly detected anomalies as a function of the number of nodes analyzed, where nodes are ranked in descending order according to the anomaly scores assigned by the model. Analogously to the ROC-AUC, we also compute the Area Under the Cumulative Gain Curve (higher is better) to facilitate a direct comparison between Janus and the strongest competitor identified in Table 2, alongside a perfect classifier (the

| Model | Disney | | Books | | Reddit | | T-Finance | | ACM | | Flickr | |
|---|---|---|---|---|---|---|---|---|---|---|---|---|
| | ROC-AUC | AP | ROC-AUC | AP | ROC-AUC | AP | ROC-AUC | AP | ROC-AUC | AP | ROC-AUC | AP |
| LOF | $0.479 \pm 0.000$ | $0.052 \pm 0.000$ | $0.365 \pm 0.000$ | $0.015 \pm 0.000$ | $0.572 \pm 0.000$ | $\mathbf{0.042 \pm 0.000}$ | $0.493 \pm 0.000$ | $0.052 \pm 0.000$ | $0.717 \pm 0.000$ | $\mathbf{0.144 \pm 0.000}$ | $0.625 \pm 0.000$ | $\mathbf{0.202 \pm 0.000}$ |
| Isolation Forest | $0.573 \pm 0.026$ | $0.094 \pm 0.029$ | $0.417 \pm 0.011$ | $0.018 \pm 0.002$ | $0.461 \pm 0.012$ | $0.028 \pm 0.001$ | $0.636 \pm 0.019$ | $0.056 \pm 0.003$ | $0.373 \pm 0.004$ | $0.027 \pm 0.000$ | $0.263 \pm 0.012$ | $0.038 \pm 0.001$ |
| ANOMALOUS | $0.518 \pm 0.000$ | $0.072 \pm 0.000$ | $0.018 \pm 0.000$ | $0.018 \pm 0.000$ | $0.469 \pm 0.022$ | $0.030 \pm 0.002$ | $0.282 \pm 0.000$ | $0.030 \pm 0.000$ | $0.536 \pm 0.000$ | $0.041 \pm 0.000$ | $0.537 \pm 0.000$ | $0.116 \pm 0.000$ |
| DOMINANT | $0.452 \pm 0.020$ | $0.080 \pm 0.026$ | $\mathbf{0.596 \pm 0.037}$ | $\mathbf{0.032 \pm 0.004}$ | $0.559 \pm 0.002$ | $0.038 \pm 0.000$ | $0.474 \pm 0.142$ | $0.043 \pm 0.011$ | $0.605 \pm 0.001$ | $0.050 \pm 0.000$ | $0.638 \pm 0.000$ | $0.085 \pm 0.000$ |
| CONAD | $0.495 \pm 0.031$ | $0.059 \pm 0.006$ | $0.461 \pm 0.005$ | $0.017 \pm 0.000$ | $0.555 \pm 0.003$ | $0.038 \pm 0.000$ | $0.345 \pm 0.008$ | $0.032 \pm 0.000$ | $0.605 \pm 0.000$ | $0.049 \pm 0.000$ | $0.641 \pm 0.000$ | $0.086 \pm 0.000$ |
| CARD | $0.512 \pm 0.015$ | $0.059 \pm 0.002$ | $0.470 \pm 0.018$ | $0.022 \pm 0.002$ | $0.533 \pm 0.016$ | $0.035 \pm 0.001$ | OOT | OOT | OOT | OOT | OOT | OOT |
| Janus | $\mathbf{0.705 \pm 0.045}$ | $\mathbf{0.211 \pm 0.119}$ | $0.567 \pm 0.024$ | $0.038 \pm 0.010$ | $\mathbf{0.609 \pm 0.031}$ | $0.043 \pm 0.003$ | $\mathbf{0.829 \pm 0.051}$ | $\mathbf{0.284 \pm 0.117}$ | $\mathbf{0.730 \pm 0.003}$ | $0.120 \pm 0.003$ | $\mathbf{0.709 \pm 0.023}$ | $\mathbf{0.215 \pm 0.042}$ |
| % Improv. | +23.0% | +124.5% | −4.8% | +18.8% | +6.5% | +2.3% | +32.4% | +337.5% | +1.8% | −16.7% | +10.6% | +6.4% |

Table 2: Performance comparison on anomaly detection benchmarks using ROC-AUC and AP. The last row (% Improv.) shows the relative improvement of Janus over the best-performing competitor for each metric.

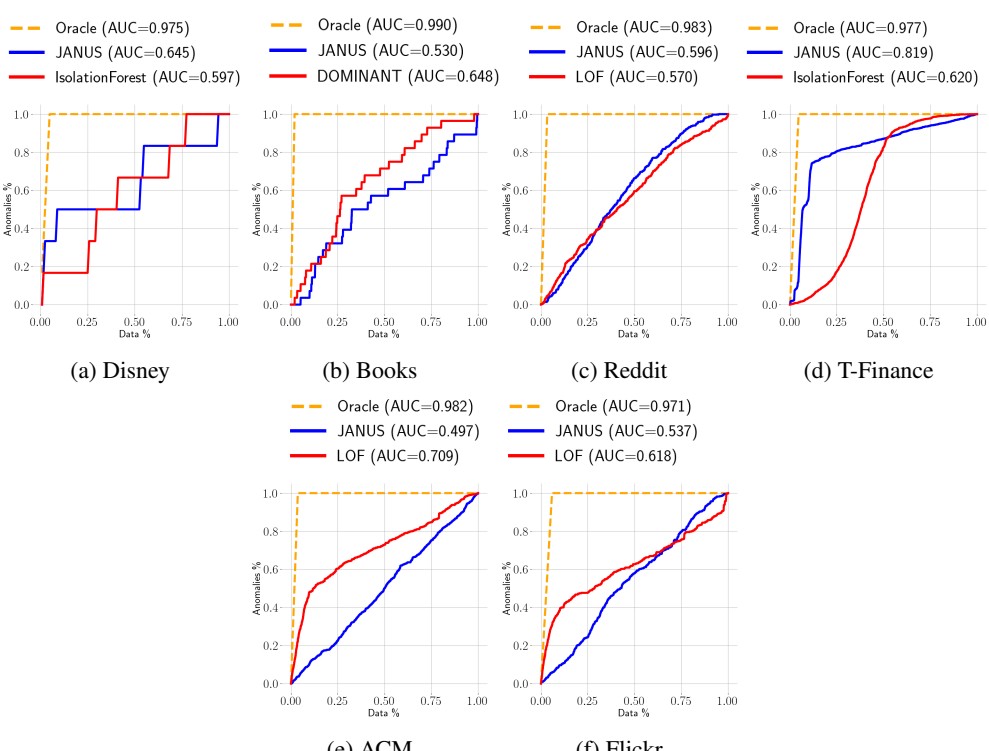

Figure 1: Cumulative Gain charts across datasets. The y-axis shows the percentage of detected anomalies, while the x-axis shows the increasing fraction of samples of the dataset adopted.

Oracle). Results empirically show that Janus consistently dominates the competitors, except for Books.

Finally, an analysis comparing the time cost of Janus against its predictive quality, in relation to the selected competitors, is presented in the Appendix A.

## 5.3 ABLATION STUDY

To address **RQ2**, Table 3 reports the results of an ablation study on the individual components of Janus. Specifically, we compare the complete model against four variants: $\text{Janus}^{AE(s, g)}$ which leverages only the autoencoder component, $\text{Janus}^{CL(s, g)}$ which relies solely on the contrastive learning component, $\text{Janus}^{E}$ and $\text{Janus}^{H}$ which retain the complete architecture but operate entirely in Euclidean or Hyperbolic space, respectively. The results highlight that Janus achieves better stability and a more balanced performance across both ROC-AUC and AP metrics.

| Model | Disney | | Books | | Reddit | | T-Finance | | ACM | | Flickr | |
|---|---|---|---|---|---|---|---|---|---|---|---|---|
| | ROC-AUC | AP | ROC-AUC | AP | ROC-AUC | AP | ROC-AUC | AP | ROC-AUC | AP | ROC-AUC | AP |
| $\text{Janus}^{AE(s,g)}$ | $0.685 \pm 0.045$ | $0.111 \pm 0.005$ | $0.644 \pm 0.004$ | $0.027 \pm 0.001$ | $0.574 \pm 0.017$ | $0.041 \pm 0.002$ | $0.789 \pm 0.055$ | $0.164 \pm 0.055$ | $0.731 \pm 0.005$ | $0.121 \pm 0.001$ | $0.730 \pm 0.021$ | $0.248 \pm 0.005$ |
| $\text{Janus}^{CL(s,g)}$ | $0.713 \pm 0.091$ | $0.183 \pm 0.054$ | $0.521 \pm 0.036$ | $0.051 \pm 0.008$ | $0.556 \pm 0.011$ | $0.039 \pm 0.001$ | $0.818 \pm 0.100$ | $0.290 \pm 0.113$ | $0.554 \pm 0.001$ | $0.043 \pm 0.005$ | $0.553 \pm 0.012$ | $0.080 \pm 0.010$ |
| $\text{Janus}^{E}$ | $0.703 \pm 0.051$ | $0.150 \pm 0.018$ | $0.555 \pm 0.029$ | $0.055 \pm 0.008$ | $0.568 \pm 0.015$ | $0.041 \pm 0.001$ | $0.694 \pm 0.143$ | $0.118 \pm 0.058$ | $0.740 \pm 0.003$ | $0.172 \pm 0.010$ | $0.740 \pm 0.021$ | $0.329 \pm 0.005$ |
| $\text{Janus}^{H}$ | $0.605 \pm 0.035$ | $0.276 \pm 0.076$ | $0.567 \pm 0.027$ | $0.057 \pm 0.009$ | $0.528 \pm 0.016$ | $0.035 \pm 0.002$ | $0.827 \pm 0.039$ | $0.222 \pm 0.075$ | $0.519 \pm 0.001$ | $0.039 \pm 0.007$ | $0.526 \pm 0.011$ | $0.067 \pm 0.010$ |
| Janus | $0.705 \pm 0.045$ | $0.211 \pm 0.119$ | $0.567 \pm 0.024$ | $0.038 \pm 0.010$ | $0.609 \pm 0.031$ | $0.043 \pm 0.003$ | $0.829 \pm 0.051$ | $0.284 \pm 0.117$ | $0.730 \pm 0.003$ | $0.120 \pm 0.003$ | $0.709 \pm 0.023$ | $0.215 \pm 0.042$ |

Table 3: Ablation results of Janus under different settings. The complete Janus achieves consistently strong performance by combining Euclidean and Hyperbolic representations.

## 5.4 DISCUSSION

Gromov's notion of $\delta$-hyperbolic space Gromov (1987) encodes features typical of the geometry of manifolds with negative curvature. Furthermore, smaller values of $\delta$ approaching zero indicates tree-like hierarchical structures (Bridson & Haefliger, 1999, Chap. III). We report the $\delta$-hyperbolic values for the adopted datasets in Table 4. The reported values align with our empirical findings: Janus performs best on the most *hyperbolic* dataset (T-Finance, $\delta$=1.0) and worse on the least *hyperbolic* (Books, $\delta$=3.0). This supports the appropriateness of introducing a hyperbolic encoder for datasets whose structure deviates from Euclidean assumptions.

| Dataset | Hyperbolicity ($\downarrow$) |
|---|---|
| Disney | 2.0 |
| Books | 3.0 |
| Reddit | 2.0 |
| T-Finance | 1.0 |
| ACM | 2.0 |
| Flickr | 1.0 |

Table 4: Hyperbolicity values of the adopted datasets

In Table 5 we report the performances of Janus (in terms of AUC and AP) by varying the parameters $\tau$ and $\lambda_1$. Notice that there is no particular trend, showing that the aforesaid parameters are dataset-dependent. In Appendix we report the full analysis including the other datasets.

| Dataset | $\tau$ | $\lambda_1$ | AUC | AP |
|---|---|---|---|---|
| | 0.3 | 0.1 | 0.609 | 0.243 |
| | 0.6 | 0.1 | 0.675 | 0.447 |
| Disney | 1.0 | 0.1 | 0.593 | 0.239 |
| | 0.6 | 0.1 | 0.675 | 0.447 |
| | 0.6 | 0.01 | 0.605 | 0.232 |
| | 0.6 | 0.001 | 0.767 | 0.413 |

Table 5: Effect of $\tau$ and $\lambda_1$ parameters on Janus performance across datasets.

## 6 CONCLUSION

In this work, we introduced Janus, a novel multi-geometry framework that integrates Euclidean and Hyperbolic representations for node-level anomaly detection. By jointly leveraging the complementary properties of both spaces within a graph autoencoder equipped with a contrastive objective, Janus captures subtle structural and semantic irregularities that single-geometry models tend to overlook. Our extensive evaluation across six real-world datasets demonstrates that Janus consistently outperforms state-of-the-art baselines, yielding significant improvements in both ROC-AUC and Average Precision.

## REPRODUCIBILITY STATEMENT

To ensure reproducibility of our results, we provide full experimental support, including the implementation of our method along with preprocessing and training scripts, available at `https://anonymous.4open.science/r/JANUS-5EDF/`. These resources allow other researchers to replicate our results and explore additional applications of our approach.

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

# A TIME ANALYSIS

Figure 2 depicts the relationship between detection accuracy, quantified through AP and ROC-AUC, and computational cost, expressed in terms of processing time (seconds), for all the considered datasets. While some baselines either achieve lower accuracy or need high execution timings, Janus demonstrates a balanced trade-off by combining competitive efficiency with consistently superior predictive performance. This highlights its practical suitability for real-world scenarios where both accuracy and scalability are crucial.

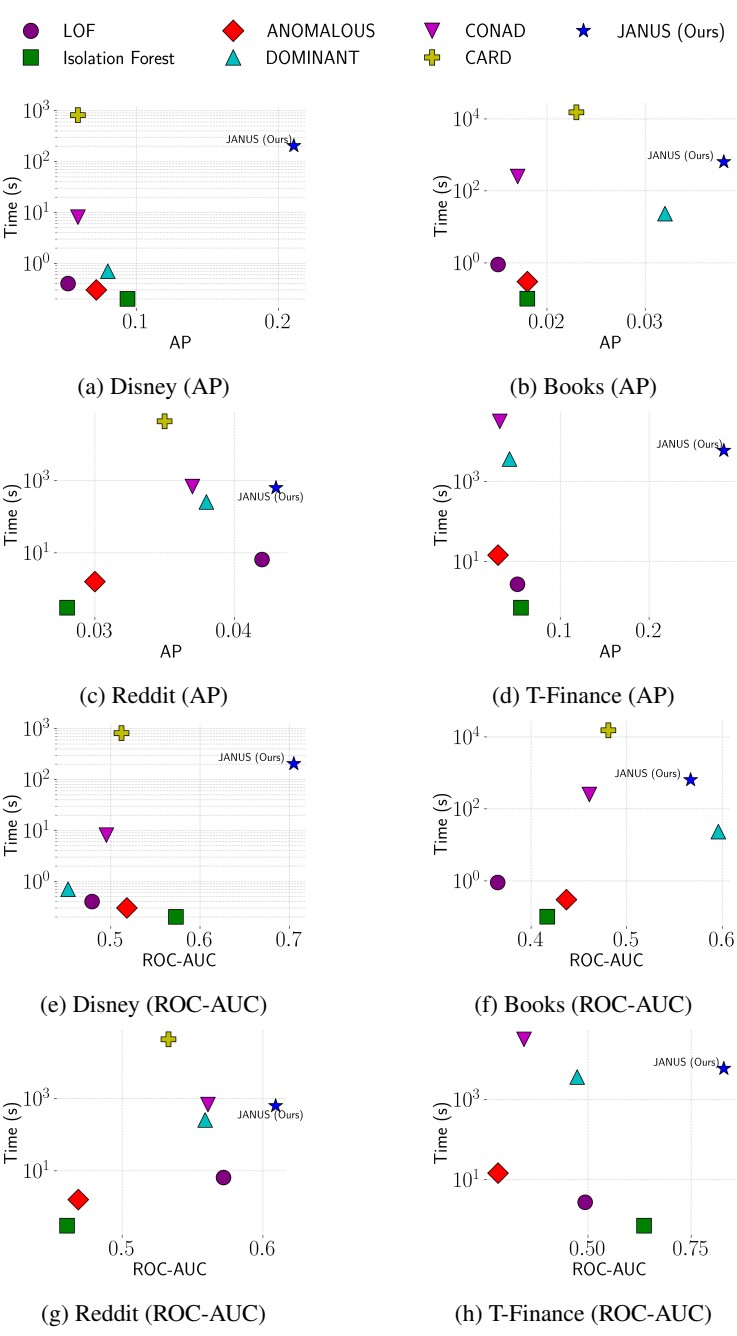

Figure 2: Comparison of models across datasets. Top row: AP vs Time. Bottom row: ROC-AUC vs Time. The legend is shared across all subplots.

## B  EXPERIMENTAL SETUP

Table 6 provides the complete list of hyperparameters employed for training Janus.

| Training Hyperparameters | | | |
|---|---|---|---|
| Learning Rate | $[0.0001, 0.001, 0.01]$ | Layers | $[3, 5]$ |
| Hidden Channels | $[8, 32]$ | $RW, DG$ | $[4, 8]$ |
| Temperature $\tau$ | $[0.3, 0.6, 1.0]$ | $\lambda_1$ | $[0.1, 0.01, 0.001]$ |
| $\lambda_2$ | $1.0$ | | |

Table 6: Hyperparameters and settings used for the Janus's training.

COMPETITOR MODELS

- **LOF** Breunig et al. (2000). Local Outlier Factor is a density-based anomaly detection method that identifies anomalies by comparing the local density of a node with those of its neighbors. Nodes residing in significantly lower-density regions are considered anomalous.

- **Isolation Forest** Liu et al. (2008). This ensemble-based method isolates anomalies by recursively partitioning the data space with random decision trees. Since anomalies are more easily separated than normal points, they require fewer splits to be isolated, making this method computationally efficient and scalable.

- **ANOMALOUS** Peng et al. (2018). A pioneering shallow graph-based method, ANOMA-LOUS applies a CUR matrix decomposition on the attribute–structure interaction matrix of the graph. The model detects abnormal nodes by capturing inconsistencies between topology and node features.

- **DOMINANT** Ding et al. (2019). A deep learning–based model that extends autoencoder architectures to graphs. DOMINANT jointly reconstructs both the adjacency matrix and node attributes via GCNs, with large reconstruction errors signaling potential anomalies.

- **CONAD** Xu et al. (2022). A contrastive learning framework designed for anomaly detection in attributed graphs. CONAD generates augmented graph views and leverages contrastive objectives to distinguish normal nodes (consistent across views) from anomalous ones (inconsistent across views).

- **CARD** Wang et al. (2024). The model refines contrastive representation learning by incorporating neighborhood-aware augmentations and anomaly-oriented regularization. Its design enables it to capture subtle irregularities in both topology and node features.

## C  SENSITIVITY ANALYSIS

| Dataset | $\tau$ | $\lambda_1$ | AUC | AP |
|---|---|---|---|---|
| Disney | 0.3 | 0.1 | 0.609 | 0.243 |
| | 0.6 | 0.1 | 0.675 | 0.447 |
| | 1.0 | 0.1 | 0.593 | 0.239 |
| | 0.6 | 0.1 | 0.675 | 0.447 |
| | 0.6 | 0.01 | 0.605 | 0.232 |
| | 0.6 | 0.001 | 0.767 | 0.413 |
| Books | 0.3 | 0.1 | 0.601 | 0.047 |
| | 0.6 | 0.1 | 0.589 | 0.054 |
| | 1.0 | 0.1 | 0.563 | 0.038 |
| | 0.6 | 0.1 | 0.589 | 0.054 |
| | 0.6 | 0.01 | 0.622 | 0.039 |
| | 0.6 | 0.001 | 0.525 | 0.043 |
| Reddit | 0.3 | 0.001 | 0.588 | 0.046 |
| | 0.6 | 0.001 | 0.659 | 0.047 |
| | 1.0 | 0.001 | 0.564 | 0.040 |
| | 0.6 | 0.1 | 0.559 | 0.039 |
| | 0.6 | 0.01 | 0.552 | 0.039 |
| | 0.6 | 0.001 | 0.659 | 0.047 |
| T-Finance | 0.3 | 0.1 | 0.582 | 0.059 |
| | 0.6 | 0.1 | 0.582 | 0.059 |
| | 1.0 | 0.1 | 0.582 | 0.059 |
| | 0.3 | 0.1 | 0.582 | 0.059 |
| | 0.3 | 0.01 | 0.584 | 0.059 |
| | 0.3 | 0.001 | 0.584 | 0.060 |
| ACM | 0.3 | 0.01 | 0.727 | 0.119 |
| | 0.6 | 0.01 | 0.697 | 0.089 |
| | 1.0 | 0.01 | 0.669 | 0.079 |
| | 0.1 | 0.1 | 0.695 | 0.086 |
| | 0.1 | 0.01 | 0.732 | 0.124 |
| | 0.1 | 0.001 | 0.710 | 0.099 |
| Flickr | 0.3 | 0.001 | 0.660 | 0.121 |
| | 0.6 | 0.001 | 0.700 | 0.162 |
| | 1.0 | 0.001 | 0.673 | 0.132 |
| | 0.1 | 0.1 | 0.727 | 0.237 |
| | 0.1 | 0.01 | 0.720 | 0.223 |
| | 0.1 | 0.001 | 0.729 | 0.260 |

Table 7: Effect of $\tau$ and $\lambda_1$ parameters on Janus performance across datasets.

