# OpenReview forum: "Combining Euclidean and Hyperbolic Representations for Node-level Anomaly Detection"
_ICLR.cc/2026/Conference — Submitted to ICLR 2026_

### Official Review · Reviewer_CrQM · 2025-10-18

**Soundness:** 2
**Presentation:** 2
**Contribution:** 2
**Rating:** 2
**Confidence:** 4

**Summary:**

The authors introduce Janus, a framework that jointly leverages Euclidean and Hyperbolic Graph Neural Networks to capture complementary aspects of node representations for unsupervised graph anomaly detection. Given experiments show the effectiveness to some extent.

**Strengths:**

1. The authors introduce a framework that jointly leverages Euclidean and Hyperbolic Graph Neural Networks for unsupervised graph anomaly detection.
2. Given experiments show the effectiveness to some extent.

**Weaknesses:**

1. The design should be further explained. For example, in the algorithm, the authors tend to use the loss as the anomly score for each node, but during the training procedure, the minimization of loss will lead to a low score for all the nodes, which makes it difficult for the framework to learn useful signals. Furthermore, directly setting the coefficient of adjacency reconstruction to 0 during inference requires reasonable explanations.
2. The compared baselines are not comprehensive enough. The authors mainly consider the old baselines in the unsupervised anomaly detection area. They should include novel baselines, such as [1-3].
3. The compared datasets are not representative enough. The included datasets can not show the effectiveness of the framework as there are sevearl real-world anomaly detection datasets in [4]. The authors should conduct further experiments on those datasets, especially the largest ones, to simulate the real deployment.
4. The authors should provide hyperparameter analysis. Without hyperparameter analysis, it is hard to see the influence of different hyperparameters.
5. The authors should explain how they chose hyperparameters. As shown in Table 4, they utilize a grid search technique for the training procedure. However, it is difficult for unsupervised learning to decide which hyperparameters should be chosen. Besides, they should also explain how they choose hyperparameters for new datasets.

[1] Hezhe Qiao, Guansong Pang. Truncated Affinity Maximization: One-class Homophily Modeling for Graph Anomaly Detection. NeurIPS 2023.

[2] Jingyan Chen, Guanghui Zhu, Chunfeng Yuan, Yihua Huang. Boosting Graph Anomaly Detection with Adaptive Message Passing. ICLR 2024.

[3] Xiangyu Dong, Xingyi Zhang, Yanni Sun, Lei Chen, Mingxuan Yuan, Sibo Wang. SmoothGNN: Smoothing-aware GNN for Unsupervised Node Anomaly Detection. WWW 2025.

[4] Jianheng Tang, Fengrui Hua, Ziqi Gao, Peilin Zhao, Jia Li. GADBench: Revisiting and Benchmarking Supervised Graph Anomaly Detection. NeurIPS 2023.

**Questions:**

Please refer to the weaknesses.

---

> ### Author Response · Authors · 2025-11-24
>
> [W1] We note that using reconstruction or contrastive losses as anomaly indicators is standard practice in unsupervised node anomaly detection (NAD) [1,2,3], where the key insight is that anomalous nodes typically exhibit higher reconstruction or misalignment errors than normal nodes. During training, the loss is minimized across all nodes; however, the optimization does not force all nodes to have identical embeddings. In fact, normal nodes converge to consistent structural-feature patterns, whereas anomalies remain misaligned. This divergence naturally produces higher anomaly scores for atypical nodes.
>
> During inference, the adjacency reconstruction component is set aside and not used for scoring. Its role during training is to stabilize the learning process by encouraging embeddings to respect local structural patterns, while the actual anomaly score relies on the cross-geometry alignment signal and node reconstruction error.
>
> [W2] We respectfully disagree that the baselines are outdated. CONAD (2022) and CARD (2024) are the most recent, widely-used unsupervised NAD methods and are included in recent PyGOD benchmark releases. Further, SmoothGNN: Smoothing-aware GNN for Unsupervised Node Anomaly Detection doesn’t provide the source code.
>
> [W3] Our experiments cover four diverse datasets with organic anomalies of varying sizes, including up to 39K nodes and 21M edges, which we believe is already representative for testing NAD frameworks. We recognize that real-world anomaly detection often involves larger-scale networks. To that end, we have extended our experiments to additional datasets (ACM and Flickr). Please see answer W6 to reviewer QoLC.
>
> [W4] We will provide a more detailed hyperparameter analysis in the revised version, showing the influence of key parameters on detection performance.
>
> [W5] We are sorry if this was not clear in the paper. We adopted the same procedure as in prior works [1, 2]: we conduct a tuning phase adopting the different hyperparameters according to Table 4, eventually taking the best configuration which maximizes the Average Precision metric. The same approach is adopted for new datasets.
>
> [1] Wang, Y., Wang, X., He, C., Chen, X., Luo, Z., Duan, L., & Zuo, J. (2024, July). Community-Guided Contrastive Learning with Anomaly-Aware Reconstruction for Anomaly Detection on Attributed Networks. In International Conference on Database Systems for Advanced Applications (pp. 199-209). Singapore: Springer Nature Singapore.
>
> [2] Fu, Y., Li, J., Liu, J., Xing, Q., Wang, Q., & King, I. (2024). Hc-glad: Dual hyperbolic contrastive learning for unsupervised graph-level anomaly detection. arXiv preprint arXiv:2407.02057.
>
> [3] Luo, Xuexiong, et al. "Deep graph level anomaly detection with contrastive learning." Scientific Reports 12.1 (2022): 19867.

---

> > ### Comment · Reviewer_CrQM · 2025-11-27
> >
> > Thanks for the rebuttal. However, the explanations can not address my concerns.
> >
> > For W1, the heuristic design of the loss can be a main question, which can not be simply addressed by the current explanation. I believe the authors should provide a case study or experiments to illustrate their points.
> >
> > For W2, I have followed the current baselines in unsupervised graph anomaly detection for some time, and I think the current works will include [1-3] in their comparison. Besides, [3] has provided their codes in https://github.com/xydong127/SmoothGNN.
> >
> > For W3, I can not understand if there are large datasets specifically designed for graph anomaly detection, which are commonly used in the related area, why the authors still conduct additional experiments on datasets designed for node classification. I believe the authors should provide experiments on Amazon, Elliptic, DGraph-Fin, and T-Social.
> >
> > For W4, could the authors point out where the added hyperparameter analysis is?
> >
> > For W5, I would like to know how this procedure addresses my concerns of hyperparameter tuning for unsupervised models, as they still need to see the performance and then choose the best one.
> >
> > [1] Hezhe Qiao, Guansong Pang. Truncated Affinity Maximization: One-class Homophily Modeling for Graph Anomaly Detection. NeurIPS 2023.
> >
> > [2] Jingyan Chen, Guanghui Zhu, Chunfeng Yuan, Yihua Huang. Boosting Graph Anomaly Detection with Adaptive Message Passing. ICLR 2024.
> >
> > [3] Xiangyu Dong, Xingyi Zhang, Yanni Sun, Lei Chen, Mingxuan Yuan, Sibo Wang. SmoothGNN: Smoothing-aware GNN for Unsupervised Node Anomaly Detection. WWW 2025.

---

### Official Review · Reviewer_QoLC · 2025-10-28

**Soundness:** 3
**Presentation:** 3
**Contribution:** 2
**Rating:** 4
**Confidence:** 4

**Summary:**

This paper proposes a node-level graph anomaly detection method called Janus. It combines a Euclidean GNN and a hyperbolic GNN with a cross-geometry contrastive regularizer. The loss combining alignment across views and feature/adjacency reconstruction is used both for training and as the anomaly score. Experiments on four real-world datasets show improvements over shallow and deep baselines. Ablation studies are presented to show contributions of both components.

**Strengths:**

1. This paper takes an interesting approach: using mixed-curvature representations to improve node-level anomaly detection. The design of using two geometries is well-motivated and shown to be effective.
2. The real-world datasets, including a large, dense finance graph, make the problem meaningful in practice.
3. The inclusion of algorithmic pseudocode and a link to the implementation enhances reproducibility and transparency.

**Weaknesses:**

1. While the combination of Euclidean and hyperbolic spaces for anomaly detection is interesting, mixed-curvature and product-space ideas have already appeared in other contexts. The anomaly detection framework is also not new. The novelty here lies primarily in applying the mixed-curvature design to the existing node-level anomaly detection framework and is thus limited.
2. Additional ablation can be done to check the contribution of each geometry, say considering Euclidean-only and hyperbolic-only variants of the model.
3. The authors do not justify why hyperbolic space is appropriate for the chosen datasets. Evidence such as hierarchical graph structures, degree distributions, or hyperbolicity analysis could better motivate the use of non-Euclidean embeddings.
4. Eq (12) is specifically for GCN, not a general GNN.
5. This paper does not have sensitivity analysis for the key hyperparameters.
6. The baseline methods seem weak for the chosen datasets. Several baseline methods yield ROC-AUC $< 0.5$ or $\sim 0.5$, implying either inadequate tuning or that the methods are not suitable for the datasets. For instance, CONAD and CARD were originally evaluated on different datasets and achieved much higher performance. To strengthen credibility, the authors should either introduce stronger baseline methods or include more datasets as presented in CONAD and CARD.

**Questions:**

1. [W2] Can you perform ablation studies to check the contribution of each geometry?
2. [W3] Why is hyperbolic space suitable for the datasets you consider?
3. [W5] Can you perform sensitivity analysis for the key hyperparameters?
4. [W6] Can you either introduce stronger baseline methods or include more datasets as presented in CONAD and CARD?

---

> ### Author Response · Authors · 2025-11-24
>
> [W1] We acknowledge that mixed-curvature and product-manifold ideas have appeared in other tasks. Our contribution is not in proposing a new geometric theory, but in demonstrating that cross-geometry alignment is particularly powerful for node-level anomaly detection (NAD), a task that is both practically critical and methodologically challenging. NAD is essential in domains ranging from cybersecurity to fraud detection and social network monitoring, where identifying nodes with subtle structural or feature irregularities can prevent cascading failures or critical security breaches. Janus is the first NAD model to jointly leverage Euclidean and hyperbolic encoders with geometry-aware contrastive regularization, enabling anomalies to be exposed precisely when structural and feature-based embeddings disagree. This approach directly addresses the dual nature of node anomalies, capturing deviations in both connectivity patterns and attribute semantics.
>
> [W2] We included ablations comparing Euclidean-only, Hyperbolic-only, and the full Janus model. The single-geometry variants consistently underperform or behave less stably, confirming that Euclidean and hyperbolic embeddings capture complementary anomaly signals. The full results are reported in response W1/Q1 to reviewer kbB6.
>
> [W3] Following the reviewer’s suggestion, we computed the Gromov δ-hyperbolicity of each dataset (lower values indicate more tree-like, hierarchical structure, better suited to hyperbolic geometry):
>
> - Disney: 2.0
> - Books: 3.0
> - Reddit: 2.0
> - T-Finance: 1.0
>
> These values align with our empirical findings: Janus performs best on the most hyperbolic dataset (T-Finance, δ=1.0) and worse on the least hyperbolic (Books, δ=3.0). This supports the appropriateness of introducing a hyperbolic encoder for datasets whose structure deviates from Euclidean assumptions.
>
> [W5] We appreciate the reviewer’s comment, and we will add a sensitivity analysis regarding tau and lambda_1 (lambda_2 is fixed).
>
> [W6] Following the reviewer’s suggestion, we added ACM and Flickr datasets adopted in CARD. Here we present the results:
>
> (ROC-AUC mean ±- std , AP mean ± std)
>
> ACM
> - LOF: 0.717 ± 0.000, 0.144 ± 0.000
> - Isolation Forest: 0.373 ± 0.004, 0.027 ± 0.000
> - ANOMALOUS: 0.536 ± 0.000, 0.041 ± 0.000
> - DOMINANT: 0.605 ± 0.001, 0.050 ± 0.000
> - CONAD: 0.605 ± 0.000, 0.049 ± 0.000
> - CARD: OOT, OOT
> - Janus: 0.73 ± 0.003, 0.120 ± 0.003
>
> Flickr
> - LOF: 0.625 ± 0.000, 0.202 ± 0.000
> - Isolation Forest: 0.263 ± 0.012, 0.038 ± 0.001
> - ANOMALOUS: 0.537 ± 0.000, 0.116 ± 0.000
> - DOMINANT: 0.638 ± 0.000, 0.085 ± 0.000
> - CONAD: 0.641 ± 0.000, 0.086 ± 0.000
> - CARD: OOT, OOT
> - Janus: 0.709 ± 0.023, 0.215 ± 0.042
>
> CARD did not finish within 12 hours on either dataset, consistent with its known computational overhead. CARD is slow because it relies on community-guided contrastive learning (requiring community detection, multiple augmentations, and large similarity computations) combined with an anomaly-aware reconstruction module requiring both encoder and decoder passes. This makes it computationally expensive on large or dense graphs.

---

### Official Review · Reviewer_kbB6 · 2025-10-31

**Soundness:** 2
**Presentation:** 1
**Contribution:** 2
**Rating:** 2
**Confidence:** 4

**Summary:**

The paper proposes the Janus framework, which claims to improve the performance of node-level anomaly detection by jointly using Euclidean and hyperbolic graph neural networks to capture complementary features of node representations. Its core design involves constructing two views of raw features and structural features for each node, embedding them in Euclidean and hyperbolic spaces respectively, and then aligning the embedding space with the graph autoencoder combined with the contrast learning object. Finally, the nodes that are difficult to coordinate with the views are determined as anomalies.

**Strengths:**

1.	This paper explores the application of “dual geometric spaces” in the NAD task, offering a novel technical approach for this field.

2.	The paper is well-structured and contains no obvious errors.

**Weaknesses:**

1. Motivation Lack of Necessity Argument: The paper does not address the core question of "why a single geometric model cannot meet the needs of NAD".

2. The baseline model selection is outdated and has not been compared with the latest methods.

3. The paper does not provide a model diagram, making it difficult to gain a clear understanding of the overall model architecture.

**Questions:**

1.	Can the author provide a stronger theoretical justification for why combining Euclidean and hyperbolic spaces is effective for anomaly detection tasks?

2.	The introduction of hyperbolic space will increase the computational cost. Please compare the parameters, training time, and inference time between Janus and the baseline model, explain the trade-off between the model's efficiency and performance, and prove its applicability in real scenarios.

---

> ### Author Response · Authors · 2025-11-24
>
> [W1/Q1] Hyperbolic spaces proved an effective tool [1] to encode the hierarchical structures of complex networks [2]. In particular, they provide a tool for embedding hierarchical structures in low dimensions, as opposed to expanding the embedding size.
> Mixed-curvature embeddings [3, 4] provide empirical grounding that combining geometries captures complementary modes of variation. Janus operationalizes this by aligning Euclidean and hyperbolic embeddings via a product-metric contrastive loss. Nodes that cannot be jointly aligned across geometries naturally receive higher anomaly scores.
> Empirically, we present an ablation study to show that Euclidean-only and Hyperbolic-only variants perform worse or less stably than the joint model, demonstrating the necessity of combining geometries:
>
> (ROC-AUC mean ±- std , AP mean ± std)
>
> Disney
> - Janus-E: 0.703 ± 0.051, 0.150 ± 0.018
> - Janus-H: 0.605 ± 0.035, 0.276 ± 0.076
> - Janus:  0.705 ± 0.045, 0.211 ± 0.119
>
> Books
> - Janus-E: 0.555 ± 0.029, 0.055 ± 0.008
> - Janus-H: 0.567 ± 0.027, 0.057 ± 0.009
> - Janus:  0.567 ± 0.024, 0.038 ± 0.010
>
> Reddit
> - Janus-E: 0.568 ± 0.015, 0.041 ± 0.001
> - Janus-H: 0.528 ± 0.016, 0.035 ± 0.002
> - Janus:  0.609 ± 0.031, 0.043 ± 0.003
>
> T-Finance
> - Janus-E: 0.694 ± 0.143, 0.118 ± 0.058
> - Janus-H: 0.827 ± 0.039, 0.222 ± 0.075
> - Janus:  0.829 ± 0.051, 0.284 ± 0.117
>
>
> [W2] We respectfully disagree that the baselines are outdated. CONAD (2022) and CARD (2024) are the most recent, widely-used unsupervised NAD methods and are included in recent PyGOD benchmark releases.
>
> [W3] We agree and will include a clear architectural diagram in the revision, illustrating the dual-view construction, Euclidean/Hyperbolic encoders, product-metric contrastive alignment, and reconstruction modules.
>
> [Q2] We already include a timing-vs-performance analysis (Appendix Fig. 2). Results show:
> - Some baselines are faster but significantly less accurate.
> - Others (e.g., CARD) are slower and fail to run on large graphs (OOT on T-Finance).
> - Janus provides the best accuracy–efficiency trade-off and reliably trains on large datasets.
>
> The hyperbolic overhead is modest: message passing occurs entirely in the tangent space (purely Euclidean operations), with only one log-map and one exp-map per batch. Parameter counts are comparable to Euclidean GNNs since the curvature is fixed and no additional geometry parameters are learned.
> On the largest dataset (39K nodes, 21M edges), Janus obtains 0.829 ROC-AUC, far above all baselines, confirming scalability and practical value.
>
> [1] Krioukov, Dmitri, et al. "Hyperbolic geometry of complex networks." Physical Review E—Statistical, Nonlinear, and Soft Matter Physics 82.3 (2010): 036106.
> [2] Clauset, Aaron, Cristopher Moore, and Mark EJ Newman. "Hierarchical structure and the prediction of missing links in networks." Nature 453.7191 (2008): 98-101.
> [3] Cho, Sungjun, et al. "Curve your attention: Mixed-curvature transformers for graph representation learning." arXiv preprint arXiv:2309.04082 (2023).
> [4] Gu, Albert, et al. "Learning mixed-curvature representations in product spaces." International conference on learning representations. 2018.

---

### Meta-Review · Area_Chair_KzjV · 2025-12-14

**Summary:**

The work introduces a new method called Janus for graph anomaly detection at the node level. The key idea is to learn AD-oriented embeddings across the Euclidean and Hyperbolic spaces. The effectiveness of Janus is evaluated on six real-world GAD datasets.

**Reviewer Concerns:**

The reviewers raised the following key concerns:
1. The paper lacks a strong motivation for why a single geometric model is insufficient for the node-level AD task.
2. The novelty is limited since mixed-curvature ideas and the anomaly detection framework already exist in prior work.
3. The model architecture is insufficiently explained, with no diagram and unclear design choices.
4. Baseline methods are outdated and/or weak, and important recent baselines are missing.
5. The paper lacks key ablations, such as Euclidean-only and hyperbolic-only variants.
6. Hyperparameter sensitivity analysis is absent.

The author rebuttal does not justify well why the mixed-curvature idea is crucial for identifying node anomalies.  Methods [1-3] pointed out by Reviewer CrQM represent state-of-the-art in various lines of GAD research, which should be compared to deliver more convincing empirical evidence. The hyperparameter settings and justification are still a key concern.

The mixed-curvature idea shows some better performance than Euclidean/hyperbolic space-based variants, but it is more an idea that applies prior mixed-curvature  approaches to NAD, without major technical innovation.

In terms of the datasets, commonly used real-world GAD datasets are missing, such as Amazon, Yelp, DGraph, etc. ACM and Flickr datasets contain artificial anomalies only, which are often less preferable than those with real anomalies.

Overall, most of the key concerns remain, and the AC agrees with the reviewers on these concerns.

**Reviewer Scores:**

There are two rejects and one weak reject. The rebuttal helps address some of the minor issues, while most of the major ones remain.

---

### Decision · Program_Chairs · 2026-01-26

Reject